# Field Tests for Assessing Functional Capacity in Children with Chronic Lung Diseases Other than Asthma: A Scoping Review

**DOI:** 10.3390/healthcare13192417

**Published:** 2025-09-24

**Authors:** Panagiotis Dalamarinis, Eleni A. Kortianou, Aspasia Mavronasou, Vaia Sapouna, Dafni Moriki, Konstantinos Douros

**Affiliations:** 1“Paediatric Respiratory Medicine”, Medical School, National and Kapodistrian University of Athens, 12462 Athens, Greece; 2Clinical Exercise Physiology and Rehabilitation Laboratory, Physiotherapy Department, School of Health Sciences, University of Thessaly, 35132 Lamia, Greece; ekortianou@uth.gr (E.A.K.); asmavronasou@uth.gr (A.M.); vsapouna@uth.gr (V.S.); 3Pediatric Allergy and Respiratory Unit, 3rd Department of Pediatrics, “Attikon” University Hospital, School of Medicine, National and Kapodistrian University of Athens, 12462 Athens, Greece; dmoriki@med.uoa.gr (D.M.); kdouros@med.uoa.gr (K.D.)

**Keywords:** 1-min sit-to-stand test, 6MWT, cystic fibrosis, functional capacity, functional test, non-cystic fibrosis bronchiectasis, pediatric, primary ciliary dyskinesia, shuttle walk test, step test

## Abstract

**Objective:** To synthesize the available evidence on field tests used to assess functional capacity in children with CLDs other than asthma, such as cystic fibrosis (CF), and non-CF bronchiectasis (NCFB). Still, the application and reliability of the field tests in non-asthmatic pediatric CLDs populations is scarce. **Methods:** Three databases (PubMed, Medline via EBSCOhost, and Web of Science) were searched from inception to 20 May 2025. Two researchers independently screened the retrieved articles and rated the methodological quality using the NIH Quality Assessment Tool for Observational Cohort and Cross-Sectional Studies. Information was extracted about study design, field test used, outcomes measured, and methodological quality. **Results:** Out of 784 records, 8 studies met the inclusion criteria. Most studies focused on CF. Five different field tests were identified: six-minute walk test (6MWT), modified shuttle walk test (mSWT), one-minute sit-to-stand test (1mSTS), three-minute step test (3mST), and TGlittre-P test. The 6MWT (*n* = 3) and mSWT (*n* = 2) were the most frequently used and demonstrated good reliability and clinical applicability. Reported outcomes included distance walked, total steps, task’ repetitions, and cardiopulmonary parameters, such as heart rate and perceived exertion of dyspnea/leg fatigue. **Conclusions:** Field exercise tests appear to be feasible in children with CLDs other than asthma, with most data available in CF. They can be used to monitor functional capacity over time, to assess the effectiveness of rehabilitation programs, and to complement symptom assessment with tools such as the Borg scale. Evidence in NCFB and PCD is still limited, and additional pediatric studies are needed.

## 1. Introduction

Chronic lung diseases (CLDs) in children encompass a spectrum of conditions characterized by structural abnormalities of the lungs and impaired pulmonary function, which may lead to significant respiratory symptoms [1]. Cystic Fibrosis (CF), non-Cystic Fibrosis Bronchiectasis (NCFB), and Primary Ciliary Dyskinesia (PCD) are commonly grouped under the term chronic suppurative lung diseases (CSLDs), a clinical phenotype characterized by chronic endobronchial infection, ongoing inflammation, and mucopurulent airway secretions [2].

Cystic fibrosis affects approximately 1 in 2500 live births in European populations [3], while NCFB presents with highly variable prevalence depending on diagnostic criteria, ranging from 0.2 to 735 per 100,000 children [4]. PCD, a rare congenital disorder of motile cilia, is estimated to occur in 1 in 20,000 individuals, although underdiagnosis is frequent due to clinical heterogeneity [5].

Persistent airway inflammation, ventilation-perfusion mismatch, and recurrent infections can impair cardiopulmonary efficiency, induce respiratory and peripheral muscle weakness, reduce daily physical activity levels, and negatively affect growth and nutritional status [6,7]. As a result, children with CLDs frequently demonstrate diminished exercise capacity, reduced muscle strength, and intermittently decreased participation in physical activities, all of which contribute to a lower health-related quality of life (HRQoL) [8].

Structured exercise training has emerged as a core therapeutic component in the management of chronic pediatric respiratory diseases. It has been shown to improve aerobic capacity, reduce dyspnea, enhance muscle strength, and promote better HRQoL [9,10]. In cystic fibrosis and other CSLDs, exercise interventions are associated with slower disease progression, improved pulmonary function, and decreased hospitalization rates [11].

Physiotherapists play a pivotal role in this process by conducting detailed assessments of functional and exercise capacity, providing crucial information for the design and progression of individualized rehabilitation programs. The European Respiratory Society emphasises that rehabilitation for non-asthmatic children, such as those with bronchiectasis, should be delivered by trained physiotherapists with pediatric expertise, and should include components such as airway clearance, strength training, aerobic exercise, and breathing strategies [12].

Although cardiopulmonary exercise testing (CPET) is considered the gold standard for evaluating exercise tolerance, field exercise tests, such as the six-minute walk test (6MWT), modified shuttle walk test (mSWT), and step tests, have been widely adopted as practical alternatives in the clinical setting [13].

These field-based exercise tests require minimal equipment, provide valid and reproducible estimates of functional capacity, and are generally well tolerated by children with moderate-to-severe chronic lung diseases, including cystic fibrosis [14,15].

Despite the increasing recognition of field tests in pediatric respiratory care, the literature lacks clarity regarding validity and reliability for assessing functional capacity in children with CLDs other than asthma.

Therefore, this scoping review aims to systematically map the existing field-based exercise tests used to assess functional capacity in children with chronic lung diseases other than asthma.

## 2. Method

### 2.1. Overview

This scoping review was structured following the Preferred Reporting Items for Systematic Reviews and Meta-Analyses Extension for Scoping Reviews (PRISMA-ScR) [16]. The methods were developed on the basis of the methodological framework for scoping review recommended by Arksey and O’Malley [17] as well as by Levac et al. [18]. The detailed PRISMA-ScR checklist is presented in Appendix A [19].

### 2.2. Research Questions

The research questions were as follows: “Which field tests are used to assess functional capacity in children with CLDs other than asthma?” and “Are those field tests valid and reliable?”

### 2.3. Eligibility Criteria

The inclusion criteria were original publications that (a) included children (aged 6–12 years) with CLDs except asthma, (b) investigated the use of field tests for assessing the functional capacity of the targeted group, (c) studies in the English language. Children with (a) asthma (b) cardiovascular, musculoskeletal, neurological diseases or cancer as a comorbidity, or (c) athletes, were excluded. CLDs that do not occur in pediatric populations, such as COPD, were excluded as they belong to the adult-onset spectrum. Abstracts, book reviews, book chapters, narrative reviews, systematic reviews, scoping reviews, grey literature, case series/reports, commentaries, letters to the editor, editorials, clinical practice guidelines, and protocols were also excluded.

### 2.4. Search Strategy

A systematic literature search (Appendix B) was performed from inception to 20 May 2025, using 3 electronic databases: PubMed, Medline (via EBSCOhost) and Web of Science, using a combination of subject headings and keywords. The search strategy was designed using keywords and MESH terms related to Lung Diseases, Obstructive, Exercise Test and Pediatrics. The concepts and key index terms used were adapted to the selected databases, and the keywords were combined using Boolean logical operators (AND and OR). Citations of the included articles and relevant systematic reviews were used for a hand search of additional eligible studies for inclusion.

### 2.5. Screening and Article Selection

All retrieved articles were imported into Rayyan (Qatar Computing Research Institute, 8CCG+J8J, Ar-Rayyan, Qatar) Web app (Version 1.6.1, Rayyan Systems, Inc., Boston, MA, USA), and duplicates were removed manually (https://www.rayyan.ai—accessed 20 May 2025). Two researchers (PD and AM) independently screened all titles, abstracts, and full texts for inclusion. In case of discrepancies at any stage of the study selection, a third researcher (EK) was consulted to make the final decision.

### 2.6. Data Extraction and Verification

Two reviewers (PD and VS) independently screened the retrieved articles and assessed the methodological quality of the included studies using the NIH Quality Assessment Tool for Observational Cohort and Cross-Sectional Studies. A template was developed to guide data extraction, including the author and year of publication, study design, field test, diagnosis, study population and sample size, purpose, and reported outcomes. All extracted data were synthesized and collated in a descriptive table summary.

### 2.7. Quality Assessment

To assess the methodological quality of the included studies, we employed the Quality Assessment Tool for Observational Cohort and Cross-Sectional Studies, developed by the National Heart, Lung, and Blood Institute (NHLBI), a part of the National Institutes of Health (NIH) [20]. This tool is specifically designed for use in studies that include non-randomized observational studies, such as cohort and cross-sectional designs.

The tool comprises 14 items that evaluate critical aspects of study design and conduct, including: clarity of the research question and study objectives, definition and representativeness of the study population, sample size justification, adequacy of participation rate, reliability and validity of exposure and outcome measures, temporal relationship between exposure and outcome, appropriateness of statistical analyses, and consideration of potential confounding variables.

Each item was assessed as “Yes”, “No”, “Cannot Determine”, “Not Reported”, or “Not Applicable”. Based on the number and nature of the criteria met, an overall quality rating was assigned to each study as *Good*, *Fair*, or *Poor*, following the guidance provided by the NHLBI.

Two reviewers (PD and VS) independently performed the quality assessments. Any discrepancies in scoring or overall ratings were resolved through discussion and consensus. When disagreement persisted, a third reviewer (EK) was consulted to reach a final decision. The results of the quality assessment were used to inform the interpretation of the findings and the strength of the evidence presented in this review.

### 2.8. Data Synthesis and Analysis

A narrative synthesis was conducted for relevant outcomes and methodological characteristics of all included studies. The results and discussion sections of all the included articles were presented to identify the field test used to assess the functional capacity of children with CLDs.

## 3. Results

### 3.1. Flow of Studies

The initial search from the 3 electronic databases yielded a total of 784 publications. After the removal of the duplicate articles and the titles and abstracts screening, a total of 8 articles were collected and assessed for eligibility. The PRISMA flowchart in Figure 1 shows the literature review strategy and the reasons for article exclusion.

### 3.2. Characteristics of the Included Studies

The characteristics of the included studies (*n* = 8) are presented in Table 1. They were published between September 2019 and April 2025 and employed two main methodological designs: cross-sectional (*n* = 6) and longitudinal (*n* = 2) design studies. The primary aims of the studies varied: some focused on establishing test-retest reliability (*n* = 3), minimal detectable changes (*n* = 2), cardiorespiratory responses (*n* = 5), or associations with physiological parameters such as respiratory muscle strength (*n* = 2), and lung function (*n* = 3).

Seven out of eight studies investigated children with CF, and one study included participants with PCD. Sample sizes ranged from 10 to 132 participants, with reported age ranges between 7 and 11 years.

### 3.3. Field Tests Used

Five different field tests employed in the assessment of functional capacity in children with CLDs: the 6MWT, mSWT, three-minute step test (3mST), one-minute sit-to-stand test (1mSTS), and the TGlittre-Pediatric test (TGlittre-P). The 6MWT emerged as the most utilized test, featured in three studies [21,23,26] (CF: *n* = 2, PCD: *n* = 1). The mSWT, used in two studies [25,27], involving children with CF. Each of the 3mST [22], TGlittre-P test [24] and 1mSTS [28] was used in a single study featuring children with CF.

### 3.4. Reported Outcomes of the Field Tests

The main outcome measure during the walking tests was the distance covered (in meters). During the 3mST, all participants completed the test at the prescribed cadence of 30 steps per minute without early termination. Children with CF demonstrated a higher mean rating of perceived exertion at test completion compared to healthy controls (HC), despite achieving a comparable step count and no significant differences in heart rate or oxygen saturation recovery profiles [22]. Performance in the 1mSTS was quantified by the total number of repetitions completed. Additionally, repetitions were normalized by body mass (kg), and total work (TW) was estimated by multiplying the repetitions by body mass [28]. For the TGlittre-P test, the main performance indicator was the total time required to complete the test [24]. Cardiorespiratory responses were common outcome measures across all field tests [21,22,23,24,25,26,27,28]. They included heart rate (HR), respiratory rate (RR), systolic blood pressure (SBP), diastolic blood pressure (DBP), and heart rate variation (VHR = HRpeak − HRbaseline). The rating of perceived exertion and sensation of dyspnea was assessed using the Borg scale. Peripheral oxygen saturation (SpO_2_%) was monitored at baseline, throughout, and upon completion of each test.

Among the eight studies, only one incorporated a structured assessment of cardiopulmonary recovery following test completion. Silva et al. (2021) [22], utilising the 3-min step test, measured recovery indices including the Borg dyspnea and fatigue, SBP after the test and up to five minutes of recovery.

### 3.5. Test-Retest Reliability

The test-retest reliability has been examined only in children diagnosed with CF. One of the three studies that employed the 6MWT demonstrated high test-retest reliability for the distance covered (6MWD), with intraclass correlation coefficients (ICC) ranging from 0.708 to 0.948 (mean ICC: 0.874), indicating strong reproducibility. Moreover, physiological and subjective outcomes—including heart rate (HR), oxygen saturation (SpO_2_), fatigue, and dyspnea—were consistently reported, further supporting the test’s responsiveness and construct validity. Similarly, for the mSWT, one study reported high reproducibility (ICC ≥ 0.80) across three timepoints. The TGlittre-P test also demonstrated excellent test-retest reliability, with an ICC of 0.849, supporting its consistency in measuring functional performance over time.

### 3.6. Methodological Quality Assessment

The methodological quality of the included observational studies was assessed using the National Institutes of Health (NIH) Quality Assessment Tool for Observational Cohort and Cross-Sectional Studies developed by the National Heart, Lung, and Blood Institute (NHLBI) [20]. This tool evaluates 14 criteria that encompass internal validity elements such as clear research objectives, defined populations, sample size justification, timing of exposure and outcome assessments, adequacy of follow-up, and adjustment for confounding. Each study was rated as ‘Good’, ‘Fair’, or ‘Poor’ quality based on the overall assessment of the 14 items of the NIH tool. The methodological quality of the studies is presented in Table 2.

## 4. Discussion

This scoping review provides a comprehensive synthesis of field tests employed to assess functional capacity in CLDs other than asthma, with a predominant focus on CF and limited representation of PCD. Studies on asthma were excluded, as functional performance can vary with symptomatology, airway control or medication use [29,30] and the mechanisms of exercise limitation and response to interventions differ among the other chronic lung diseases [31]. We focused on children aged 6–12 years, as in this middle childhood stage, they are cognitively able to perform functional tests reliably, while still physiologically homogeneous (prepuberty), ensuring that outcomes reflect fitness rather than pubertal variability (e.g., variable changes in height and weight) [32,33]. Furthermore, data for walking tests in the youngest children (under the age of 6) are unlikely to reflect their capabilities [34]. While field tests have long been integral to adult pulmonary rehabilitation—particularly in populations with chronic obstructive pulmonary disease (COPD)—their application in pediatric populations remains underdeveloped [35].

Among the tests identified, the 6MWT emerged as the most frequently used and well-supported field test. Its widespread adoption is likely due to its simplicity, minimal equipment requirements, and strong correlation with clinically meaningful outcomes such as forced expiratory volume in one second (FEV_1_), aerobic capacity (VO_2_peak), and physical activity levels in children with CF [36]. Its reproducibility and responsiveness to clinical changes support its utility in the clinical setting. Preceding studies in children with moderate-to-severe asthma showed that the 6MWD is significantly lower compared to healthy reference values. This diminished functional capacity likely reflects a combination of ventilatory limitation, deconditioning, and exercise-induced dynamic hyperinflation, all of which may contribute to the impaired heart rate recovery as reported in published studies among children with other CLDs such as CF [36].

The mSWT addresses an incremental format that approximates maximal effort and correlates more strongly with VO_2_peak than the 6MWT in pediatric populations [37]. Studies included in this review demonstrated good test–retest reliability, responsiveness, and the ability to distinguish between healthy peers and pediatric patients for both functional tests [25,27]. These findings align with those of del Corral et al. [38], who reported excellent test–retest reliability (ICC = 0.975) for the mSWT in children and adolescents aged 7–15 years with CF. This comparison highlights both the robustness of mSWT in CF and the need for similar estimates across other pediatric CLDs and specific age groups.

Emerging tools such as the 1mSTS and the TGlittre-P test offer additional dimensions of assessment. The 1mSTS is increasingly recognized for its relationship with lower-limb muscle strength, lung function, and CPET outcomes in children with CF [39]. Furthermore, the test offers a multidimensional assessment of functional status by simulating activities of daily living as it captures endurance, strength, balance, and coordination in a single tool. Evidence from pediatric CF cohorts supports its reliability and clinical applicability particularly in highlighting limitations that may not be evident on more aerobic-focused tests such as the 6MWT [40].

The 3mST, although less extensively studied, has emerged as a feasible and reproducible tool for evaluating submaximal exercise capacity in children with CF, both in face-to-face and remote assessment [41]. Notably, the test provides insights into the perceived exertion of dyspnea and fatigue as an indicator of reduced exercise tolerance, despite central cardiac responses (HR) that might be comparable to those of HC.

The sensation of breathlessness, as captured via ratings of RPE during functional tests, is measurable and clinically informative in children with chronic disease. In cystic fibrosis, Borg dyspnea scores obtained immediately after the 6MWT correlate with 6MWD and cardiorespiratory strain, supporting RPE as a valid symptom-limited marker of capacity [42]. Consistent with this, pediatric cardiology guidance explicitly integrates Borg dyspnea/fatigue ratings into 6MWT procedures for congenital heart disease, underscoring their relevance to recovery profiling [43].

Beyond their role in functional assessment, field tests may also serve as sensitive markers of clinical stability. Exercise tolerance, as measured by the 6MWT, is associated with a higher risk of hospitalization and earlier lung function decline in pubertal children with CF [11]. A greater 6-min walk distance (6MWD) was independently associated with a lower risk of first hospitalization and fewer total hospital days over 5 years for children with CF [44]. Similarly, reduced distance on the mSWT predicted increased hospitalization risk over 2 years, and more hospital days [45]. Field tests are also sensitive to therapeutic responses. For example, in hospitalized children and adolescents with CF, the mSWT distance improved by approximately 102 m during antibiotic and supportive therapy [46]. However, direct evidence linking field test performance to adverse outcomes in PCD or NCFB remains lacking, highlighting the need for multicenter pediatric studies in these diseases. Moreover, incorporating symptom perception measures, such as the Borg dyspnea and fatigue scales, enhances the value of field tests, capturing aspects of exercise intolerance that are not reflected by physiological indices alone [43]. In children with CF, elevated Borg scores for dyspnea or fatigue at a given workload have been linked to a greater risk of exacerbations and hospitalizations [47]. Thus, abnormal Borg scale responses may serve as early warning signals even in patients whose walking distance or sit-to-stand performance appears within the expected range in the same population [47,48]. These insights highlight that field-based assessments should not be viewed merely as substitutes for cardiopulmonary exercise testing, but rather as multidimensional tools that integrate physiological, functional, and patient-reported outcomes in clinical care.

Field tests also hold value in the rehabilitation process of children with CLDs. In pediatric CF, randomized controlled trials have demonstrated that structured exercise training leads to significant improvements in functional capacity as measured by the 6MWT [10]. Additionally, systematic reviews on pediatric CLDs confirm consistent gains in walking distance, aerobic fitness, and quality of life across exercise interventions [9]. Moreover, the 1-min sit-to-stand test has shown sensitivity to training-induced changes and strong associations with muscle strength [49]. Lastly, telerehabilitation programs incorporate field tests for monitoring progress in children diagnosed with CF [39].

### 4.1. Strengths and Limitations

A strength of this scoping review is its approach, summarizing the available evidence for a variety of functional field tests, such as walking, stepping, and multi-task tests, used in pediatric populations (6–12 years old) with chronic lung diseases, other than asthma. The review highlighted that the majority of the functional tests are used in children with CF. There is a lack of use of those tests in the clinical and research setting for populations other than CF. Importantly, by focusing on non-asthma CLDs, this review addresses an underrepresented yet clinically important pediatric subgroup.

However, several limitations should be acknowledged. The heterogeneity of included studies in terms of disease diagnosis and testing protocols limits direct assumptions. For example, the 3mST and the 1mSTS lack test-retest reliability in this specific age group. In addition, most available evidence originates from CF studies, whereas data for other conditions such as NCFB and PCD remain limited.

### 4.2. Future Recommendations

Future perspectives could aim to use the standardized protocols for field tests in pediatric CLDs other than asthma populations, enabling cross-study comparisons and reference values across tests. Multicenter studies, particularly in NCFB are needed to validate the psychometric properties and clinical responsiveness of field tests. The integration of symptom-based measures, such as sensation of dyspnea and ratings of perceived exertion, alongside physiological outcomes, could provide a more complete picture of functional limitations and patient experience. Longitudinal studies assessing the prognostic value of these field tests for clinical outcomes, including exacerbations, hospitalization risk, and quality of life, would strengthen the evidence base for their routine use in pediatric respiratory care. While evidence apart from cystic fibrosis remains limited, future research should address this gap in populations with NCFB and PCD.

## 5. Conclusions

This scoping review synthesised the available evidence on field tests for assessing functional capacity in children aged 6–12 years with CLDs other than asthma. Τhe novelty of this study lies in its focus on a specific age range of children with chronic lung diseases, such as those with CF and PCD. To the best of our knowledge, no previous studies have synthesized different types of functional exercise tests (e.g., shuttle, stepping test, and multi-task) within this particular age group and disease population. The 6MWT and the mSWT were the most commonly used tests, especially in children with CF, demonstrating good reproducibility, feasibility, and clinical usefulness. Other tools, such as the 1mSTS, the 3mST, and the TGlittre-P test, offer complementary dimensions of functional capacity. However, their feasibility and reproducibility remain limited.

## Figures and Tables

**Figure 1 healthcare-13-02417-f001:**
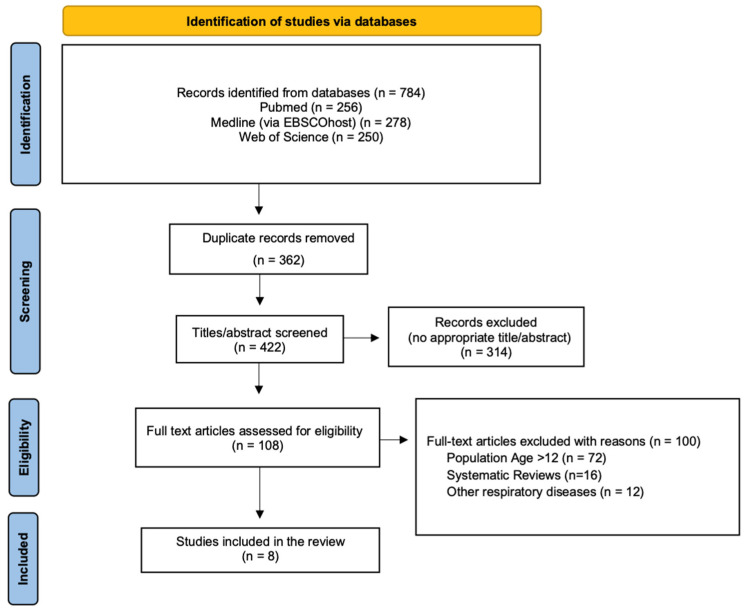
PRISMA Flow Chart.

**Table 1 healthcare-13-02417-t001:** Characteristics of the included studies.

Authors (Year)	Study Design	Field Test	Diagnosis (Sample Size) Age	Purpose	Reported Outcomes
Villanueva et al. (2019) [21]	Repeated measures concordance	6MWT	CF (*n* = 20) 8.3 ± 1.08	Test-retest reliability and establish the minimal detectable change	6MWD, Dyspnea (Borg), Fatigue (Borg), SpO_2_, HR
Silva et al. (2021) [22]	Cross-sectional study	3mST	CF (*n* = 10) 10.4 ± 3.13	Differences in cardiorespiratory fitness between children with and without CF	SBP, DBP, HR, SpO_2_, Dyspnea (Borg), Fatigue (Borg)
Innocenti et al. (2021) [23]	Multicenter cross-sectional observational study	6MWT	CF (*n* = 132) 8.7 ± 1.6	Assess physical performance and correlation between the 6MWD and respiratory function	6MWD, HR, SpO_2_, Dyspnea (VAS), Fatigue (VAS)
Scalco et al. (2021) [24]	Cross-sectional and comparative study	TGlittre-P	CF (*n* = 18) 9.5 ± 1.8	Test-retest reliability & establish the minimal detectable change	Time Spent, HR, RR, SpO_2_, Dyspnea (Borg)
Leite et al. (2021) [25]	Prospective, longitudinal study	mSWT	CF (n= 48) 10.1 ± 2.7	Assess reproducibility over a 9-month period	6MWD, Shuttle Level, HR, RR, DBP, SBP, VO_2_max, Dyspnea (Borg), VHR
Firat et al. (2022) [26]	Cross-sectional, single-center study	6MWT	PCD (*n* = 29) 10.74 ± 4.01	Comparisons between PCD and HC	6MWD, HR, RR, SpO_2_, DBP, SBP, Dyspnea (Borg), Fatigue (Borg)
Mucha et al. (2023) [27]	Cross-sectional study	mSWT	CF (*n* = 31) 10.2 ± 2.1	Compare cardiorespiratory responses between CF and HC	DA, Shuttle Level, HR, SpO_2_, DBP, SBP, Dyspnea (Borg), Fatigue (Borg)
Santos Costa et al. (2025) [28]	Observational, cross-sectional study	1mSTS	CF (*n* = 17) 9.8 ± 1.6	Assess physiological responses	Number of Repetitions, TW, HR, SpO_2_, Dyspnea (Borg), Fatigue (Borg)

Abbreviations: 1mSTS, 1-min Sit To Stand Test; 3mST, 3-min Step Test; 6MWD, 6-min Walk Distance; 6MWT, 6-min Walk Test; CF, Cystic Fibrosis; DA, Distance Achieved; DBP, Diastolic Blood Pressure; HC, Healthy Controls; HR, Heart Rate; ICC, Intraclass Correlation Coefficient; mSWT, modified Shuttle Walk Test; PCD, Primary Ciliary Dyskinesia; RR, Respiratory Rate; RPE, Rating Perceived Exertion; SBP, Systolic Blood Pressure; SpO_2_, % Oxygen Saturation; TW, Total Work; VHR, Variation of Heart Rate.

**Table 2 healthcare-13-02417-t002:** National Institutes of Health (NIH) Quality Assessment Tool for Observational Cohort and Cross-Sectional Studies.

No.	Villanueva et al. (2019) [21]	Silva et al. (2021) [22]	Innocenti et al. (2021) [23]	Scalco et al. (2021) [24]	Leite et al. (2021) [25]	Firat et al. (2022) [26]	Mucha et al. (2023) [27]	Santos Costa et al. (2025) [28]
1. Was the research question or objective clearly stated?	Yes	Yes	Yes	Yes	Yes	Yes	Yes	Yes
2. Was the study population clearly specified and defined?	Yes	Yes	Yes	Yes	Yes	Yes	Yes	Yes
3. Was the participation rate of eligible persons at least 50%?	Yes	Yes	Yes	Yes	Yes	Yes	Yes	Yes
4. Were all subjects recruited from similar populations?	Yes	Yes	Yes	Yes	Yes	Yes	Yes	Yes
5. Was a sample size justification, power description, or variance and effect estimates provided?	No	No	Yes	No	Yes	No	No	No
6. Were the exposure(s) measured prior to the outcome(s)?	N/A	N/A	N/A	N/A	N/A	N/A	N/A	N/A
7. Was the timeframe sufficient to detect an association?	Yes	Yes	Yes	Yes	Yes	Yes	Yes	Yes
8. Were different levels of the exposure of interest examined?	N/A	N/A	N/A	N/A	N/A	N/A	N/A	N/A
9. Were the exposure measures clearly defined and reliable?	Yes	Yes	Yes	Yes	Yes	Yes	Yes	Yes
10. Was the exposure assessed more than once over time?	N/A	N/A	N/A	N/A	N/A	N/A	N/A	N/A
11. Were the outcome measures valid and reliable?	Yes	Yes	Yes	Yes	Yes	Yes	Yes	Yes
12. Were outcome assessors blinded?	N/R	N/R	N/R	N/R	N/R	N/R	N/R	N/R
13. Was loss to follow-up ≤20%?	N/A	N/A	N/A	N/A	No	N/A	N/A	N/A
14. Were confounding variables measured and adjusted?	Yes	Yes	Yes	Yes	Yes	Yes	Yes	Yes
Rate	Fair	Fair	Good	Fair	Fair	Fair	Fair	Fair

Abbreviations: No, Criteria Number; N/A, not applicable; N/R, not reported.

## Data Availability

No new data were created or analyzed in this study. Data sharing is not applicable to this article.

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
