# Peer review of "Field Tests for Assessing Functional Capacity in Children with Chronic Lung Diseases Other than Asthma: A Scoping Review"

_healthcare, 2025, doi:10.3390/healthcare13192417_

Round 1
Reviewer 1 Report
Comments and Suggestions for Authors
Thank you for the opportunity to review this paper.
The aim of this review is to provide a systematic view of field-based exercise tests used to assess functional capacity of children (age 6-12 yrs) with CLD other than asthma.
Despite the heterogenity of the included studies, the results of this study provide important conclusions for clinical practice. The methods are comprehensively described, and, based on the references, the role of the 6MWT and mSWT was confirmed for the functional assessment of this study population.
Minor comments:
-Based on the results of this sudy, the authors reveal the role of the field exercise tests as markers of clinical stability (line 269). Please add a comment about the role of this tests in the follow-up of this patients with non-asthma CLD, (eventually recommmandations or guidelines) similar with that for patients with congenital heart disease . Have they any role in the evaluation of the efficacy of the therapeutic agents used? Also, this study shows that the results of the fileld-tests are predictors of adverse outcome. Basd on these results, is possible to define a risk-stratification for these patients?
-Based on the results of this study, please add a comment about the eventually role of the field-tests in the rehabilitation process of these patients
- This study reveal the role of the Borg dyspnea/fatigue scales that enhance the value of the field-tests. Please add more data and comments about the role of this combination in the prevention of the future major events, he visits to the ED or the need for hospitalization.
Author Response
Response to Reviewer 1 Comments
|
||
Summary |
|
|
We sincerely thank the reviewer for his/her time and effort in reviewing our manuscript. Please find the detailed responses below and the corresponding revisions/corrections here in italics and in red font in the re-submitted manuscript.
|
||
Point-by-point response to Comments and Suggestions for Authors |
||
Comments 1: Based on the results of this study, the authors reveal the role of the field exercise tests as markers of clinical stability (line 269). Please add a comment about the role of this tests in the follow-up of this patients with non-asthma CLD, (eventually recommmandations or guidelines) similar with that for patients with congenital heart disease. Have they any role in the evaluation of the efficacy of the therapeutic agents used? Also, this study shows that the results of the fileld-tests are predictors of adverse outcome. Based on these results, is possible to define a risk-stratification for these patients?
|
||
Response 1: We thank the reviewer for this important comment, which allowed us to expand the discussion on the clinical applicability of field exercise tests. The manuscript has as follows:
Pages 9, lines 274 to 286: “Beyond their role in functional assessment, field tests may also serve as sensitive markers of clinical stability. Exercise tolerance, as measured by the 6MWT, is associated with a higher risk of hospitalization and earlier lung function decline in pubertal children with CF [11]. A greater 6-minute walk distance (6MWD) was independently associated with a lower risk of first hospitalization and fewer total hospital days over 5 years for children with CF [43]. Similarly, reduced distance on the mSWT predicted increased hospitalization risk over 2 years, and more hospital days [44]. Field tests are also sensitive to therapeutic responses. For example, in hospitalized children and adolescents with CF, the mSWT distance improved by approximately 102 m during antibiotic and supportive therapy [45]. However, direct evidence linking field test performance to adverse outcomes in PCD or NCFB remains lacking, highlighting the need for multicenter pediatric studies in these diseases.”
|
||
Comments 2: Based on the results of this study, please add a comment about the eventually role of the field-tests in the rehabilitation process of these patients
|
||
Response 2: We thank the reviewer for this valuable suggestion. We agree that beyond their diagnostic and prognostic roles, field exercise tests may have an important role in pulmonary rehabilitation for children with chronic lung diseases (CLDs). In our revised manuscript, we added the following paragraph:
Page 10, lines 295 to 302: “Field tests also hold value in the rehabilitation process of children with CLDs. In pediatric CF, randomized controlled trials have demonstrated that structured exercise training leads to significant improvements in functional capacity as measured by the 6MWT [10]. Additionally, systematic reviews on pediatric CLDs confirm consistent gains in walking distance, aerobic fitness, and quality of life across exercise interventions [48]. Moreover, the 1-min sit-to-stand test has shown sensitivity to training-induced changes and strong associations with muscle strength [49]. Lastly, telerehabilitation programs incorporate field tests for monitoring progress in children diagnosed with CF [38].
|
||
Comments 3: This study reveal the role of the Borg dyspnea/fatigue scales that enhance the value of the field-tests. Please add more data and comments about the role of this combination in the prevention of the future major events, he visits to the ED or the need for hospitalization.
|
||
Response 3: We thank the reviewer for this comment. In our revised manuscript, we expanded the Discussion to highlight the prognostic and preventive potential of this combined approach as follows:
Page 10, lines 287 to 291: “In children with CF, elevated Borg scores for dyspnea or fatigue at a given workload have been linked to a greater risk of exacerbations and hospitalizations [46]. Thus, abnormal Borg scale responses may serve as early warning signals even in patients whose walking distance or sit-to-stand performance appears within the expected range in the same population [46, 47].”
|
Reviewer 2 Report
Comments and Suggestions for Authors
The authors have wonderfully addressed a clinically relevant topic. This is a scoping review that synthesizes the available field tests used to assess functional capacity in children (6–12 years) with chronic lung diseases (CLDs) other than asthma. The authors searched three databases (PubMed, Medline, Web of Science) and identified eight eligible studies. The review highlights that the six-minute walk test (6MWT) and modified shuttle walk test (mSWT) were the most commonly used and best validated. In addition, one-minute sit-to-stand test (1mSTS), three-minute step test (3mST), and TGlittre-Pediatric test were less studied and require further validation.
However, the key takeaway from the review is missing. The conclusion should be stronger and clinical implications should be stated. Some sentences are very long and some are repeated across different sections.
Abstract:
Line 28: “most frequently used”. Quantify the number of studies in which they are used.
Conclusion: Repetition of sentences from the result. Make it stronger.
Introduction:
Line 39: First sentence has 2 “lead to”. Rewrite it for clarity.
Line 80: Why is asthma excluded? Is it because there are many studies related to asthma?
Line 81: Why is a scoping review done instead of a systematic review?
Methods:
Line 95: Why are the children of the only age group, 6-12 years old, included?
Line 111: Correct it to hand search.
Line 160: It is stated that articles were published between 2019-2025. State the month as well. Check if articles have been published after May 2025 and include them.
Results:
Line 216: Clarify “collective strength of its design and reporting”.
HC, as a healthy control, is not defined before.
Discussion:
Line 233: Since it is a review paper, change “we observed” to “we observed in published studies” or “previous studies have reported” or something related.
Line 244-245: The study group is CLDs excluding asthma. So, what does that mean for that population?
Line 290: Why sparse data? What is missing?
Conclusion:
Line 304: Start by emphasizing the novelty of this review.
Line 309: Clarify “insufficiently validated”.
The last sentence of the conclusion can be stated in the future research section.
Comments on the Quality of English LanguageSome sentences are long and should be shortened for clarity. Remove repeated sentences across different sections of the review paper.
Author Response
Response to Reviewer 2 Comments
|
||
Summary |
|
|
We sincerely thank the reviewer for his/her time and effort in reviewing our manuscript. Please find the detailed responses below and the corresponding revisions/corrections here in italics and in red font in the re-submitted manuscript.
|
||
Point-by-point response to Comments and Suggestions for Authors |
||
Comments 1: Line 28: “most frequently used”. Quantify the number of studies in which they are used.
|
||
Response 1: We appreciate the reviewer's important observation. We now pointed it out in our revised manuscript as follows:
Page 1, lines 27 to 29: “The 6MWT (n=3) and mSWT (n=2) were the most frequently used and demonstrated good reliability and clinical applicability”.
|
||
Comments 2: Conclusion: Repetition of sentences from the result. Make it stronger.
|
||
Response 2: We revised the Conclusion to avoid repeating results and to highlight the broader implications of our findings. Therefore, in our revised manuscript, the conclusion is as follows:
Page 1, lines 32 to 36: “Field exercise tests appear to be feasible in children with CLDs other than asthma, with most data available in CF. They can be used to monitor functional capacity over time, to assess the effectiveness of rehabilitation programs, and to complement symptom assessment with tools such as the Borg scale. Evidence in NCFB and PCD is still limited, and additional pediatric studies are needed.”
|
||
Comments 3: Line 39: First sentence has 2 ‘lead to’. Rewrite it for clarity.
|
||
Response 3: The sentence has been revised for clarity:
Page 3, lines 41 to 43: “Chronic lung diseases (CLDs) in children encompass a spectrum of conditions characterized by structural abnormalities of the lungs and impaired pulmonary function, which may lead to significant respiratory symptoms [1].”
|
||
Comments 4: Line 80: Why is asthma excluded? Is it because there are many studies related to asthma?
|
||
Response 4: We thank the reviewer for this important note. Asthma was excluded because field exercise tests have been extensively investigated (GINA 2024). By excluding asthma, we aimed to map the available functional tools that are used for diseases with progressive, irreversible lung structural abnormalities that are less responsive to pharmacological treatment. Therefore, in our revised manuscript, we modified the first paragraph of the Discussion as follows:
Page 8, lines 224 to 227: “Studies on asthma were excluded, as functional performance can vary with symptomatology, airway control or medication use [28, 29] and the mechanisms of exercise limitation and response to interventions differ among the other chronic lung diseases [30].”
Comments 5: Line 81: Why is a scoping review done instead of a systematic review?
Response 5: We would like to clarify to the reviewer that the primary research question examined within the present review did not constitute the primal question of published studies (Munn et al., 2018). Therefore, we have conducted a thorough and transparent research for relevant studies and have followed best practices conducting and reporting a scoping review, including clearly outlining the eligibility criteria for studies (Tricco et al., 2016).
Munn, Z.; Peters, M.D.J.; Stern, C.; Tufanaru, C.; McArthur, A.; Aromataris, E. Systematic review or scoping review? Guidance for authors when choosing between a systematic or scoping review approach. BMC Med. Res. Methodol. 2018, 18, 143. https://doi.org/10.1186/s12874-018-0611-x.
Tricco, A.C.; Lillie, E.; Zarin, W.; et al. A scoping review on the conduct and reporting of scoping reviews. BMC Med. Res. Methodol. 2016, 16, 15. https://doi.org/10.1186/s12874-016-0116-4.
|
||
Comments 6: Why are the children of the only age group, 6-12 years old, included?
Response 6: We appreciate the reviewer’s question. The choice of the 6–12 years age range reflects both developmental considerations and methodological feasibility. This middle childhood stage (6-12years) is a pivotal period of development for physical fitness and cognitive development. In this stage of age, children are cognitively able to perform functional tests reliably, while still physiologically homogeneous (prepuberty), ensuring that outcomes reflect fitness rather than pubertal variability (e.g. variable changes in height and weight) Furthermore, data for walking tests in the youngest children (under the age of 6) are unlikely to reflect their capabilities.
Therefore, in the section of the discussion in our revised manuscript, we added the following sentences:
Page 8, lines 227 to 232: “We focused on children aged 6–12 years, as in this middle childhood stage, they are cognitively able to perform functional tests reliably, while still physiologically homogeneous (prepuberty), ensuring that outcomes reflect fitness rather than pubertal variability (e.g. variable changes in height and weight) [31, 32]. Furthermore, data for walking tests in the youngest children (under the age of 6) are unlikely to reflect their capabilities [33].”
Comments 7: Line 111: Correct it to hand search. Response 7: The term has been corrected to “hand search” for accuracy.
Page 3, lines 112-113: “Citations of the included articles and relevant systematic reviews were used for a hand search of additional eligible studies for inclusion.”
|
||
Comments 8: Line 160: It is stated that articles were published between 2019-2025. State the month as well. Check if articles have been published after May 2025 and include them. Response 8: Following your suggestion, we checked if they are any articles published after May 25 till September 6, 2025, and no further eligible studies were found. We have stated months between 2019-2025, and now the manuscript has as follows:
Page 5, lines 162-164: “They were published between September 2019 and April 2025 and employed two main methodological designs: cross-sectional (n = 6) and longitudinal (n = 2) design studies”
Comments 9: Line 216: Clarify “collective strength of its design and reporting”. Response 9: We agree that the phrase required clarification. The term “collective strength of its design and reporting” refers to the overall methodological strength of each study, as determined by the National Institutes of Health (NIH) Quality Assessment Tool for Observational Cohort and Cross-Sectional Studies. Therefore, in our revised manuscript, the sentence has as follows:
Page 8, lines 218 to 219: “Each study was rated as ‘Good’, ‘Fair’, or ‘Poor’ quality based on the overall assessment of the 14 items of the NIH tool.”
Comments 10: HC, as a healthy control, is not defined before. Response 10: We appreciate the reviewer’s observation. We have now defined the abbreviation HC (Healthy Controls) at its first mention in the Results section to ensure clarity.
Page 7, lines 183 to 186: “Children with CF demonstrated a higher mean rating of perceived exertion at test completion compared to healthy controls (HC), despite achieving a comparable step count and no significant differences in heart rate or oxygen saturation recovery profiles [21].”
Comments 11: Line 233: Since it is a review paper, change “we observed” to “we observed in published studies” or “previous studies have reported” or something related. Response 11: To avoid implying that data were directly collected in this review, we have revised the wording to attribute findings appropriately to the original studies.
Page 9, lines 242 to 245: “This diminished functional capacity likely reflects a combination of ventilatory limitation, deconditioning, and exercise-induced dynamic hyperinflation, all of which may contribute to the impaired heart rate recovery as reported in published studies among children with other CLDs such as CF [35].”
Comments 12: Line 244-245: The study group is CLDs excluding asthma. So, what does that mean for that population? Response 12: We agree that this sentence concerning asthma confused the reviewer. Therefore, we removed the whole sentence from the revised manuscript.
Comments 13: Line 290: Why sparse data? What is missing? Response 13: We have revised the sentence to clarify what is lacking in the literature. Therefore, in our revised manuscript, we modified the sentence as follows:
Page 10, lines 315 to 316: “In addition, most available evidence originates from CF studies, whereas data for other conditions such as NCFB and PCD remain limited.”
|
Comments 14: Line 304: Start by emphasizing the novelty of this review.
Response 14: We thank the reviewer for this valuable suggestion. We have revised the Conclusion to emphasize the novelty of our review. The revised sentences have as follows:
Page 11, lines 331 to 336: “This scoping review synthesised the available evidence on field tests for assessing functional capacity in children aged 6-12 years with CLDs other than asthma. Τhe novelty of this study lies in its focus on a specific age range of children with chronic lung diseases, such as those with CF and PCD. To the best of our knowledge, no previous studies have synthesized different types of functional exercise tests (e.g., shuttle, stepping test, and multi-task) within this particular age group and disease population.”
Comments 15: Line 309: Clarify “insufficiently validated”
Response 15: We thank the reviewer for this important point. Using the phrase “insufficiently validated,” we refer to the fact that certain field tests (such as the 1mSTS, 3mST, and TGlittre-P test) have been investigated in only one or two small studies in pediatric populations with CF. There are no studies in NCFB or PCD.
Page 11, lines 338 to 340: “Other tools, such as the 1mSTS, the 3mST, and the TGlittre-P test, offer complementary dimensions of functional capacity. However, their feasibility and reproducibility remain limited.”
Comments 16: The last sentence of the conclusion can be stated in the future research section.
Response 16: We appreciate the reviewer’s suggestion. We agree and have relocated the final sentence regarding the lack of evidence outside CF to the “Future Recommendations” section.
Page 10, lines 325 to 329: “Longitudinal studies assessing the prognostic value of these field tests for clinical outcomes, including exacerbations, hospitalization risk, and quality of life, would strengthen the evidence base for their routine use in pediatric respiratory care. While evidence apart from cystic fibrosis remains limited, future research should address this gap in populations with NCFB and PCD.”
Reviewer 3 Report
Comments and Suggestions for Authors
This manuscript reviews the different functional tests for non-asthmatic chronic lung diseases in children of obesity
I offer the following observations for the authors:
Major points -
- Line 280 – “…systematically synthesizing…” Given the narrative/descriptive nature, even though due to the heterogenous nature of the research findings, it may be more apt to avoid making bolder claims of systematic synthesis, and instead use a subtle tone acknowledging the limitations
- Table 2 does not specify the questions used for critical appraisal.
- Moreover, it is not clear how so many questions could be answered given the different nature of cross-sectional studies and those with follow-up
- The lack of a reporting checklist raises questions on adherence to the same
- Lines 160 – 162 : Four different types of study design seems odd. Aren’t they just cross-sectional (5 + 1), and longitudinal (including the multicentric one)
- Line 144 mentions strength of evidence being assessed but this is nowhere to be seen in the report
- As per recommendations, the systematic search strategy should be documented and made available to ensure research findings are transparent. This is especially important for such reviews
- The manuscript feels more like a pile of findings rather than a narration being weaved, or at the least a coherent image being drawn, or even being summarised
- There is no mention of skipping a key chronic lung disease, i.e., COPD
Comments on the Quality of English Language
Several places throughout the manuscript, starting from the first paragraph of the abstract where the two sentences do not seem to be linked appropriately
The manuscript needs drastic changes
Author Response
Response to Reviewer 3 Comments
|
||
Summary |
|
|
We sincerely thank the reviewer for his/her time and effort in reviewing our manuscript. Please find the detailed responses below and the corresponding revisions/corrections here in italics and in red font in the re-submitted manuscript.
|
||
Point-by-point response to Comments and Suggestions for Authors
|
||
Comments 1: Line 280 – “…systematically synthesizing…” Given the narrative/descriptive nature, even though due to the heterogenous nature of the research findings, it may be more apt to avoid making bolder claims of systematic synthesis, and instead use a subtle tone acknowledging the limitations
|
||
Response 1: We acknowledge this important issue that the reviewer highlights. Therefore, we revised the sentence as follows:
Page 10, lines 305 to 310: “A strength of this scoping review is its approach, summarizing the available evidence for a variety of functional field tests, such as walking, stepping, and multi-task tests, used in pediatric populations (6-12 years old) with chronic lung diseases, other than asthma. The review highlighted that the majority of the functional tests are used in children with CF. There is a lack of use of those tests in the clinical and research setting for populations other than CF.”
|
||
Comments 2: Table 2 does not specify the questions used for critical appraisal.
|
||
Response 2: We appreciate the reviewer's observation. We have now revised Table 2 to include the 14 critical appraisal questions from the NIH Quality Assessment Tool for Observational Cohort and Cross-Sectional Studies, ensuring clarity and transparency in how the studies were assessed.
|
||
Comments 3: Moreover, it is not clear how so many questions could be answered given the different nature of cross-sectional studies and those with follow-up The lack of a reporting checklist raises questions on adherence to the same |
||
Response 3: To clarify the issue raised by the reviewer, we point out that in lines 209-215 (page 8) in our first manuscript we mentioned that the tool we used is a Quality Assessment Tool for Observational Cohort and Cross-Sectional Studies. To avoid confusion, we added the 14 criteria of the tool, and now they are clearly presented in Table 2 of our revised manuscript.
|
||
Comments 4: Lines 160 – 162: Four different types of study design seems odd. Aren’t they just cross-sectional (5 + 1), and longitudinal (including the multicentric one)
|
||
Response 4: We thank the reviewer for this helpful suggestion. We agree that our initial phrasing was unnecessarily complex. We have simplified the description of study designs by classifying them into cross-sectional (including the multicenter study) and longitudinal designs, which provides greater clarity and consistency. Therefore in our revised manuscript, the sentence has as follows:
Page 5, lines 162 to 164: “They were published between September 2019 and April 2025 and employed two main methodological designs: cross-sectional (n = 6) and longitudinal (n = 2) design studies.” |
||
Comments 5: Line 144 mentions strength of evidence being assessed but this is nowhere to be seen in the report Response 5: We apologize for this fault. We agree that our wording was confusing. In this scoping review, we assessed the methodological quality of individual studies using the NIH Quality Assessment Tool. We have revised the text to clarify that only methodological quality was evaluated, not the strength of evidence. Therefore, we revised the text as follows:
Page 3, lines 121 to 123: “Two reviewers (PD and VS) independently screened the retrieved articles and assessed the methodological quality of the included studies using the NIH Quality Assessment Tool for Observational Cohort and Cross-Sectional Studies”
Comments 6: As per recommendations, the systematic search strategy should be documented and made available to ensure research findings are transparent. This is especially important for such reviews.
The systematic search strategy is presented in Appendix A as reported in Page 3, line 104 of our first manuscript. To further clarify the strategy process of the findings, we added the full list of search terms and Boolean combinations applied across PubMed, Web of Science, and MEDLINE (EBSCOhost) in the revised Appendix A.
Comments 7: There is no mention of skipping a key chronic lung disease, i.e., COPD Response 6: We would like to clarify that COPD was not skipped; rather, it was not relevant to the scope of our review, which was strictly limited to children aged 6–12 years. As COPD is an adult-onset disease and does not occur in pediatric populations, it was not considered among the conditions included or excluded. |

Round 2
Reviewer 3 Report
Comments and Suggestions for Authors
Regarding the comment 3, the authors may clarify how loss to follow-up etc. was addressed in the risk of bias assessment
Moreover, a reporting checklist may be used
Original comment 3:
“Moreover, it is not clear how so many questions could be answered given the different nature of cross-sectional studies and those with follow-up
The lack of a reporting checklist raises questions on adherence to the same”
______
The authors have answered the seventh comment, and a target is that they address this upfront in the manuscript itself for the benefit of the uninitiated reader; I have attached the seventh response below:
“We would like to clarify that COPD was not skipped; rather, it was not relevant to the scope of our review, which was strictly limited to children aged 6–12 years. As COPD is an adult-onset disease and does not occur in pediatric populations, it was not considered among the conditions included or excluded”
Author Response
Response to Round 2 Comments
|
||
Summary |
|
|
We sincerely thank the reviewer for their time and effort in reviewing our manuscript. Please find the detailed responses below and the corresponding revisions/corrections here in italics and in red font in the re-submitted manuscript.
|
||
Point-by-point response to Comments and Suggestions for Authors |
||
Comments 1: Regarding the comment 3, the authors may clarify how loss to follow-up etc. was addressed in the risk of bias assessment Moreover, a reporting checklist may be used Original comment 3: The lack of a reporting checklist raises questions on adherence to the same”
Response 1: We thank the reviewer for this insightful revision. We agree that certain items of the NIH Quality Assessment Tool are not equally applicable to all study designs. Criterion that assesses the risk of bias due to loss to follow-up (criterion 13 of the NIH Quality Assessment Tool) is inherently relevant to cohort or longitudinal studies but not to cross-sectional studies. We regret that we did not clarify this, in our Table 2. Thus, we marked this criterion for all the studies as “Yes” in our first manuscript. To minimize potential confusion, we have revised Table 2 so that for cross-sectional studies, “loss to follow-up” (criterion 13) is now explicitly indicated as “N/A” or “No” rather than “Yes”.
As reposted in our manuscript (Section 2.1, Overview, pages 2-3), this review was structured in accordance with the PRISMA-ScR framework. Therefore, a PRISMA-ScR reporting checklist is presented in Table S1 of our revised manuscript.
|
||
Comments 2: The authors have answered the seventh comment, and a target is that they address this upfront in the manuscript itself for the benefit of the uninitiated reader; I have attached the seventh response below: “We would like to clarify that COPD was not skipped; rather, it was not relevant to the scope of our review, which was strictly limited to children aged 6–12 years. As COPD is an adult-onset disease and does not occur in pediatric populations, it was not considered among the conditions included or excluded”
Response 2: We thank the reviewer for this suggestion. We added the following sentence in our revised manuscript:
Page 3, lines 102-103: “CLDs that do not occur in pediatric populations, such as COPD, were excluded as they belong to the adult-onset spectrum.”
|